# Measuring uncertainty in human visual segmentation

**Jonathan Vacher** [1¤]*, **Claire Launay**[2], **Pascal Mamassian**[1☙], **Ruben Coen-Cagli** [2,3,4☙]*

**1** Laboratoire des systèmes perceptifs, Département d'études cognitives, École normale supérieure, PSL University, CNRS, Paris, France, **2** Department of Systems and Computational Biology, Albert Einstein College of Medicine, Bronx, New-York, United States of America, **3** Dominick P. Purpura Department of Neuroscience, Albert Einstein College of Medicine, Bronx, New-York, United States of America, **4** Department of Ophthalmology and Visual Sciences, Albert Einstein College of Medicine, Bronx, New-York, United States of America

☙ These authors contributed equally to this work.
¤Current address: Université Paris Cité, CNRS, MAP5, Paris, France
* jonathan.vacher@u-paris.fr (JV); ruben.coen-cagli@einsteinmed.edu (RC-C)

## Abstract

Segmenting visual stimuli into distinct groups of features and visual objects is central to visual function. Classical psychophysical methods have helped uncover many rules of human perceptual segmentation, and recent progress in machine learning has produced successful algorithms. Yet, the computational logic of human segmentation remains unclear, partially because we lack well-controlled paradigms to measure perceptual segmentation maps and compare models quantitatively. Here we propose a new, integrated approach: given an image, we measure multiple pixel-based same–different judgments and perform model–based reconstruction of the underlying segmentation map. The reconstruction is robust to several experimental manipulations and captures the variability of individual participants. We demonstrate the validity of the approach on human segmentation of natural images and composite textures. We show that image uncertainty affects measured human variability, and it influences how participants weigh different visual features. Because any putative segmentation algorithm can be inserted to perform the reconstruction, our paradigm affords quantitative tests of theories of perception as well as new benchmarks for segmentation algorithms.

**Data Availability Statement:** The vseg package is available at link: https://vseg.gitlab.io/vseg/ All human data are available on Zenodo at link: https://doi.org/10.5281/zenodo.8101695.

## Author summary

Visual segmentation is the process of decomposing the visual field into meaningful parts. Segmentation is the focus of a vast literature in visual perception and neuroscience, because it is a core function of the visual system that involves bottom/up and top/down integration across the whole visual cortex. Similarly, segmentation is an essential task of computer vision systems, because it is required for countless practical applications. However, the lack of rigorous empirical measures of segmentation-related uncertainty represents a major roadblock for both fields, because subjective uncertainty is a central feature of visual perception, and also because existing databases do not allow to calibrate

**Funding:** RCC is supported by NIH grants EY031166 and EY030578. PM and JV are supported by ANR grants ANR-19-NEUC-0003-01 and ANR-17-EURE-0017. The funders had no role in study design, data collection and analysis, decision to publish, or preparation of the manuscript.

**Competing interests:** The authors have declared that no competing interests exist.

segmentation algorithms that do compute uncertainty. The work presented in this manuscript proposes to overcome these limitations. Specifically, our contributions are three-fold: (i) We introduce the first experimental method to measure perceptual segmentation on arbitrary images. (ii) We capture individual-level variability and relate it to perceptual uncertainty, which is necessary to understand human perception. (iii) We offer computational tools to fit any segmentation algorithm to the data, which will enable new benchmarks for computer vision algorithms, and testing computational theories of perceptual segmentation.

## Introduction

The processes of segmenting a visual scene into individual objects and grouping elementary visual features to build those objects, are central to visual perception [1], and therefore have been addressed extensively in both vision research [1–8] and artificial intelligence [9].

Thanks to progress in machine learning, the field of image segmentation in computer vision has flourished in the past decades. Modern algorithms achieve high performance in engineering applications ranging from general purpose segmentation of natural scenes [10–13] and scene understanding [14, 15], to medical image analysis [16] and animal pose estimation [17]. Besides their practical success, these algorithmic frameworks offer a promising toolbox to support scientific inquiry of human perceptual grouping and segmentation [18–24]. This is analogous to deep learning architectures for object recognition, which currently provide the most accurate identification of objects in natural images and movies, possibly mimicking neural processes in primate visual cortex [25–27]. Yet, the current experimental paradigms to measure perceptual grouping and segmentation are still very basic, and they fall short of providing a sufficiently detailed representation that would be necessary for a quantitative understanding of the algorithmic bases of those perceptual processes [28].

We can identify at least three shortcomings of existing human segmentation databases of natural images, that have been used to train machine learning algorithms [29–33]. First, these databases invariably rely on manual tracing of the contours of visual groups, but do not control for interactions between perceptual processes and motor planning and execution that can introduce bias and variability, neither of which reflects perceptual processing per se. Specifically, smooth tracing movements require less effort than discontinuous movements, therefore participants may be biased to segment the image using smoother boundaries than what they perceive. Furthermore, this effect can translate into variability between different participants, because their effort level is also likely to vary. Second, typically there are no constraints on, nor measurements of timing, thereby introducing additional uncontrolled variability across participants. Third, even though some databases include segmentation maps produced by multiple participants for the same image, and thus allow an analysis of variability across participants, existing databases do not measure the variability of the segmentation map produced by an individual participant. This is a crucial shortcoming when one considers perception as probabilistic inference to extract meaning from uncertain sensory inputs [34–36]. As we emphasize below, segmentation is a quintessential example of inference on uncertain inputs [37] because the pixels of an image often do not contain sufficient information for unequivocally labeling them as grouped or segmented. And in turn, sensory uncertainty leads to intra-individual variability, namely variability in the perceptual reports by the same individual across repeated presentations of an image, so it is important to document and model this variability.

The lack of methods that address these shortcomings is surprising because perceptual grouping and segmentation have been studied for decades with traditional visual psychophysics paradigms that do worry about these criteria [38]. However, these experiments often rely on artificial visual stimuli that are manipulated along just a few dimensions defined by the experimenter, such as the color and size of simple geometric shapes [39–41] or the orientation and spatial frequency of visual textures [37, 42–45]. Typically, the participants are asked to make same/different judgments, in order to study how simple stimulus manipulations influence the perceived groupings. This work has provided a solid foundation for our understanding of perceptual grouping [1]. For instance, this work has revealed universal Gestalt rules such as proximity, similarity and good continuation [1]; it has shown strong interactions with higher level processes such as object recognition [46–49]; and it has revealed that human perception of groups relies on near-optimal integration of multiple visual cues [50, 51]. However, this approach explains how specific objects or features are represented, but it does not provide segmentation maps of full images. This limits the applicability to natural images, because controlled manipulations of natural images are difficult to design and to interpret. In addition, this approach limits the practical value of the obtained data for training segmentation algorithms.

To address these shortcomings, we present a new experimental protocol to measure perceptual segmentation maps of arbitrary images. Our approach builds on a version of a same/different task traditionally used in psychophysics [52], and extends it to extract full segmentation maps while satisfying all the criteria listed above. To achieve this, we formulate mathematically the problem of reconstructing a segmentation map from multiple same/different measurements. We then derive numerical optimization methods to perform the reconstruction from finite data, and validate them extensively on both synthetic and real experiments. On top of reconstructing the segmentation maps, our approach brings two important advances. First, our formulation rests on probabilistic segmentation maps, namely it assumes that participants evaluate the probability that each location in the image belongs to any segment. We demonstrate that our approach offers accurate reconstructions of these probabilistic segmentation maps, thereby providing a quantification of the perceptual uncertainty involved in grouping and segmentation. In particular, by manipulating synthetic compound textures, we show that the perceptual uncertainty of human participants tracks the overall intrinsic image uncertainty, and is concentrated near texture boundaries. Second, we provide reconstruction code to fit the data with any parametric model (deterministic or probabilistic) that predicts either the underlying segmentation maps or the measured same/different judgments. We show these features on our empirical data, where we find that the participants correctly weigh different orientation channels, and that their weight profile further reflects image uncertainty. This aspect of our method enables systematic, quantitative comparison of multiple models on the same data and with the same cost function. Our code, the `vseg` package https://vseg.gitlab.io/vseg/ implemented in python using PyTorch, can thus form the basis for benchmarking diverse algorithms and theories of perceptual grouping and segmentation.

## Materials and methods

### Ethics statement

This study was conducted in accordance with the Declaration of Helsinki and was approved by the Internal Review Board of Albert Einstein College of Medicine and Montefiore Medical Center (IRB number: 2019–10297). Participants gave online signed consent to participate in

the experiment, and upon completion of the experiment they were compensated in accordance with institutional guidelines.

We first present the experimental procedure to measure the same/different judgments of human participants who were instructed to segment the image either into a predefined number of segments, or freely. We then explain how we reconstruct the segmentation maps from the same/different judgments. For this reconstruction, we highlight the important practical constraints (*e.g.* on the minimal number of trials), and we provide expressions for the loss functions involved in the reconstruction problem. We also propose a regularization method to robustly recover the segmentation maps and explain how to perform reconstruction based on different parametric models.

## Experimental procedure

All the experiments presented in this paper were conducted online on naive participants. At the beginning of an experimental session, the screen displays some text instructing the participant to partition the image in $K$ segments and some additional text to precisely define "partition" and "segments" (see S1 Video). We also conducted separate experiments where we did not specify the value of $K$, and instead instructed the participants to freely partition the image into segments.

After performing a few practice trials, the participants started the main experiment, which is divided in $N_b$ blocks of $N_t$ trials (criteria to choose $N_b$ and $N_t$ are discussed in the following sections, and specific values are provided below). At the beginning of each block, an image to be partitioned is presented on the screen for 3 seconds during which participants are free to visually explore and decide the segmentation of the image. Then the experiment proper starts. On each trial, two points on the image are selected, and participants report whether the two points belong to the same segment or not. Each point corresponds to the center of an element of a predefined grid of size $N \times N$ (this grid covers the image but is coarser than the pixel grid and $N \geqslant 3$). The two elements of the grid are selected pseudo-randomly. First, two small red circles at the selected locations are shown in isolation on a gray background for 300 ms. Immediately afterwards, the two same circles are superimposed on the to-be-segmented image for 300 ms (those durations could be different and reduced when the experiment is conducted in the lab). Thereafter, the image and the points disappear, and a response screen is presented prompting participants to report whether the two cued locations belonged to the same segment or not (Fig 1 bottom-left). The response screen remains visible until the participant reports their choice with a key press, which triggers the beginning of the next trial.

## Experimental participants

Adult participants were recruited on the online platform Prolific (www.prolific.co). From this website, they were redirected to our experiment page produced with jsPsych 6.3 (www.jspsych. org/6.3/, [53]). Then, after calibrating the size of images to be shown on the screen of the participants by estimating their viewing distance [54] and correcting for their monitor gamma [55], they started to perform the experiment as described above.

In the experiments of involving artificial texture stimuli, we recruited 30 participants in total. They were divided in two groups of 15 participants, and each group performed the experiment on a different image.

In the experiments involving natural image stimuli, we recruited 64 participants. We collected data for 8 different natural images, and data for each image were collected over 8 sessions (as explained above).

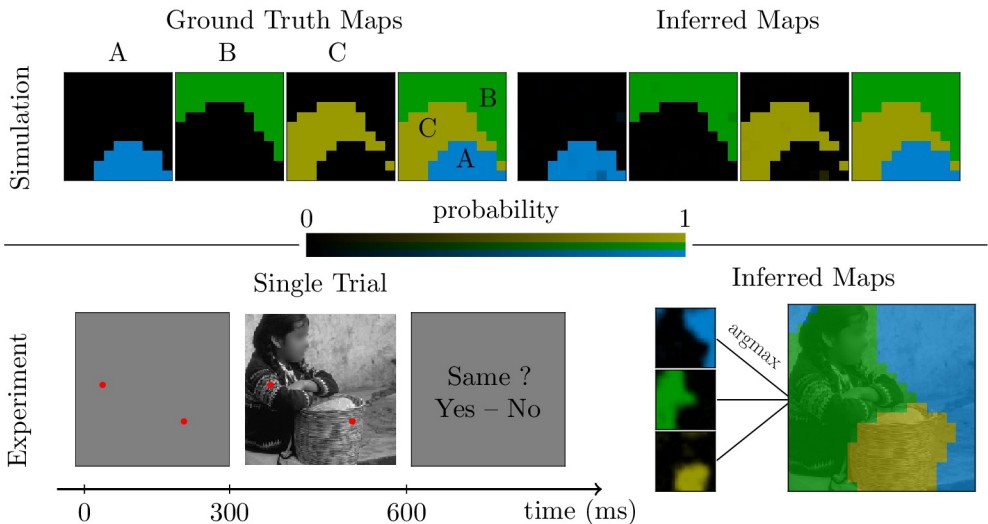

**Fig 1. Inference of segmentation maps from pairwise same/different judgments.** Top: Reconstruction of a deterministic segmentation map from simulated data (simulation details in section *Materials and methods*, subsection *Implementation and algorithm*). The leftmost panel shows the ground-truth probability map, namely the probability that each pixel belongs to the segment labeled 'A' (blue), and similarly for the second (segment 'B', green) and third (segment 'C', yellow) panel. The fourth panel from the left shows the full segmentation map, namely, for each pixel, the label of the segment with the highest probability. The four panels on the right show the corresponding maps reconstructed with the numerical procedure described in section *Materials and methods*, subsection *Inference of probabilistic segments*. Bottom-left: outline of a trial of the segmentation experiment: the participant reports whether the two locations indicated by the red dots belong to the same segment. Bottom-right: for one participant, the reconstructed probability maps (left) and corresponding segmentation map (right), obtained using spatial regularization (see section Materials and methods, subsection *Spatial regularization*).

## Stimuli

For the experiments with natural images, we used cropped natural images from the database BSD500 [29]. For the experiments involving artificial texture stimuli, we used composite textures as follows. Stimuli are images divided in two random areas which are filled with two different (but close) bandpass Gaussian noise textures (oriented textures). Image synthesis is achieved by convolving a white Gaussian noise image with a spatially-dependent filter giving the desired spectral content in each areas. Additional details are in Appendix B in S1 Text.

## Detailed choices for each experiment

The minimal number of trials $N_t$ that is necessary for the reconstruction of the segmentation map of an image is related to the grid size $N$ and the expected number of segments in the image $K$, and is precisely $N_t = (K - 1)N^2$. The explanation is given in section *Inference of probabilistic segments* paragraph *Choosing the tested pairs*. Here, we report the numbers that were used for each experiment during the development of the proposed method.

In the example experimental session of Fig 1 (bottom), the number of segments $K$ was fixed to 3. We used $N_b = 1$ and grid size $N = 19$.

In the psychophysical experiments involving artificial texture stimuli, $K$ was fixed to 2. We used $N_b = 5$, a grid size $N = 11$. We collected $N_t = KN^2 = 242$ trials (notice that this is more than the strict minimum, $N_t = (K - 1)N^2$). The median duration of each trial was 1.33 s (95% c.i. [0.98, 1.69]), including the presentation time (600 ms) and the median reaction

time. The median total duration of the experiment was approximately 40 minutes to measure the segmentation map of one image for one participant. Notice that this is substantially longer than the time during which participants were engaged with the task (median time is 27 minutes), because it includes voluntary breaks that are notoriously difficult to control in an online setting. The analyses presented in section *Results* were performed by reconstructing the segmentation maps and probability maps of each individual participant (shown in Fig E in S1 Text), and summarized in the main figures as the average maps and inferred features.

In the experiments involving natural image stimuli, $K$ was not constrained. We used $N_b = 1$ and a grid of size $N = 16$, and we collected the minimal number of trials needed to reconstruct up to $K = 5$ segments, that is we collected responses to $N_t = (K-1)N^2 = 1024$ trials. In these experiments, to limit the duration of each sessions, we divided the number of trials by 8 and collected responses to 128 trials per session, thus completing one image along 8 experimental sessions. The participants completed a session in approximately 30 minutes, including voluntary breaks. Therefore, the maps for each image were reconstructed from the aggregate data of 8 participants, not from an individual participant. See section Discussion for further considerations on the duration of the experimental session.

In all the simulations $K$ was fixed to 3 except where noted. Other simulation parameters were varied as detailed in Results.

## Inference of probabilistic segments

Given an image, a number of segments $K$, and the participant's responses, our goal is to reconstruct both the segmentation map and $K$ probability maps. Probability maps are maps that assign the probability that each pixel belongs to each of the $K$ segments, where $K \geqslant 2$. The segmentation map assigns, for each pixel, the label of the segment with the highest probability. These maps are defined on a grid of size $N \times N$ with $N \geqslant 3$. Intuitively, this requires finding the maps that are most consistent with the set of $N_b N_t$ binary responses from the participant. In turn, this involves relating the participant's judgments about whether two pixels belong to the same segment or not, to the probability that each pixel belongs to one of the $K$ segments. In this section we explain how to perform the reconstruction while treating the probability values at each pixel as free parameters. Then in the section *Parametric models*, we describe two approaches to parametrize the maps more concisely.

Formally, we use the notation $\mathcal{I}$ for the grid, *i.e.* the set of $(x, y)$ coordinates of the centers of all the elements of the grid (each element is a square, if the image length and width are equal, as in all our experiments). We use the notation $\mathcal{I}^2$ for the set of the coordinates of all the pairs of points $((x_1, y_1)$ and $(x_2, y_2))$. At each block $n \in \{1, \ldots, N_b\}$, we denote by $\mathcal{P}_n$ the set of unordered tested pairs of dots presented in each trial (*i.e.* a pair and its symmetric pair count as a single element). Note that we include in each block multiple distinct pairs, and the notation $\mathcal{P}_n$ includes the possibility that the set of pairs tested in each block is different (we will discuss further below how to optimize the choice of the pairs). Because each pair is distinct from all other pairs in the same block, the variability of the responses of one participant can only be assessed by running multiple blocks. The response of a participant at block $n$ and for a pair of pixels $(\mathbf{i}, \mathbf{j}) = ((i_x, i_y), (j_x, j_y)) \in \mathcal{I}^2$ is denoted $r_{\mathbf{i},\mathbf{j}}^{(n)}$ (note that it will be uninformative to test pairs of identical points, therefore in our experiments we exclude such pairs). We assume that participant responses $r_{\mathbf{i},\mathbf{j}}^{(n)} \in \{0, 1\}$ are independent samples of a Bernoulli random variable $R_{\mathbf{i},\mathbf{j}}^{(n)} \sim \mathcal{B}(p_{\mathbf{i},\mathbf{j}})$, with $p_{\mathbf{i},\mathbf{j}}$ denoting the probability that pixels $(\mathbf{i}, \mathbf{j})$ are perceived as belonging to the same segment. The negative log-likelihood of the dataset

$$\mathcal{D}_{N_b} = \left\{ \left( r_{\mathbf{i},\mathbf{j}}^{(n)} \right)_{(\mathbf{i},\mathbf{j}) \in \mathcal{P}_n} \right\}_{n \in \{1,\dots,N_b\}} \text{ is}$$

$$\ell_0((p_{\mathbf{i},\mathbf{j}})_{(\mathbf{i},\mathbf{j}) \in \mathcal{I}^2}; \mathcal{D}_{N_b}) = \sum_{n=1}^{N_b} \sum_{(\mathbf{i},\mathbf{j}) \in \mathcal{P}_n} \mathrm{BCE}(r_{\mathbf{i},\mathbf{j}}^{(n)} | p_{\mathbf{i},\mathbf{j}}) \tag{1}$$

where, BCE is the Binary Cross-Entropy or simply the negative log-likelihood of a Bernoulli sample $r$ (the participant's response) knowing the parameter $p$ *i.e.* $\mathrm{BCE}(r|p) = -r \log(p) - (1 - r) \log(1 - p)$. Next, because our ultimate goal is to estimate segmentation maps, we need to relate this negative log-likelihood to individual pixels rather than pairs of pixels.

In our setting, an image is assumed to have $K$ segments, and a pixel $\mathbf{i}$ belongs to segment $k$ with probability $p_{\mathbf{i}}[k] \in [0, 1]$. We also assume that the assignment of a pixel to a segment is independent of the assignments of the other pixels, given the probabilities for all the pixels (i.e. conditionally on $(p_{\mathbf{i}})_{\mathbf{i} \in \mathcal{I}}$). Under these assumptions, the probability $p_{\mathbf{i},\mathbf{j}}$ that two pixels $(\mathbf{i}, \mathbf{j})$ belong to the same segment is given by

$$p_{\mathbf{i},\mathbf{j}} = p_{\mathbf{i}} \cdot p_{\mathbf{j}} = \sum_{k=1}^{K} p_{\mathbf{i}}[k] p_{\mathbf{j}}[k] \tag{2}$$

where $p_{\mathbf{i}} = (p_{\mathbf{i}}[1], \dots, p_{\mathbf{i}}[K]) \in \Delta_K$ (the $K$-dimensional simplex), and $p_{\mathbf{i}} \cdot p_{\mathbf{j}}$ denotes the dot product. The collection $(p_{\mathbf{i}})_{\mathbf{i} \in \mathcal{I}}$ is called probabilistic segmentation maps. Therefore, by plugging Eq (2) into Eq (1) the negative log-likelihood with parametrization given by Eq (2) is

$$\ell((p_{\mathbf{i}})_{\mathbf{i} \in \mathcal{I}}; \mathcal{D}_{N_b}) = \ell_0((p_{\mathbf{i}} \cdot p_{\mathbf{j}})_{(\mathbf{i},\mathbf{j}) \in \mathcal{I}^2}; \mathcal{D}_{N_b}). \tag{3}$$

The probabilistic maps $(p_{\mathbf{i}})_{\mathbf{i} \in \mathcal{I}}$ can be estimated by minimizing the negative log-likelihood

$$(\hat{p}_{\mathbf{i}})_{\mathbf{i} \in \mathcal{I}} = \operatorname*{argmin}_{(p_{\mathbf{i}})_{\mathbf{i} \in \mathcal{I}}} \ell\left((p_{\mathbf{i}})_{\mathbf{i} \in \mathcal{I}}; \mathcal{D}_{N_b}\right) \tag{4}$$

under the constraints

$$\forall \mathbf{i} \in \mathcal{I}, \quad \sum_{k=1}^{K} p_{\mathbf{i}}[k] = 1 \ \text{ and } \ p_{\mathbf{i}} \in [0, 1]^K. \tag{5}$$

Eq (4), as for other clustering methods such as K-means or mixture models, is invariant to label permutation. Therefore, the labels found when solving the problem of Eq (4) will depend on the solver and its initialization. It is well-known that $\ell_0$ is minimized when the probability $p_{\mathbf{i},\mathbf{j}}$ is equal to the empirical mean of the responses $(r_{\mathbf{i},\mathbf{j}}^{(n)})_n$. As for Generalized Linear Models (GLMs) [56], it is worth knowing under which conditions $\ell$ is minimized when the probability $p_{\mathbf{i}} \cdot p_{\mathbf{j}}$ is equal to the empirical mean of the responses $(r_{\mathbf{i},\mathbf{j}}^{(n)})_n$. The answer is given by the following proposition.

**Proposition 1**. *Suppose that for all tested pixels $\mathbf{i} \in \mathcal{I}$ the family $(p_{\mathbf{j}})_{\mathbf{j}|(\mathbf{i},\mathbf{j}) \in \mathcal{P}}$ is a sub-family of critical point of $\ell$ and of $\ell_s$ and is linearly independent ($\mathbf{j}|(\mathbf{i}, \mathbf{j}) \in \mathcal{P}$ reads "$\mathbf{j}$ such that $(\mathbf{i}, \mathbf{j})$ belongs to $\mathcal{P}$"). Then, the optimization problem (4) is equivalent to the following least square optimization*

$$(\hat{p}_{\mathbf{i}})_{\mathbf{i} \in \mathcal{I}} = \operatorname*{argmin}_{(p_{\mathbf{i}})_{\mathbf{i} \in \mathcal{I}}} \ell_s\left((p_{\mathbf{i}})_{\mathbf{i} \in \mathcal{I}}; \mathcal{D}_{N_b}\right) \ \text{ where } \ \ell_s((p_{\mathbf{i}})_{\mathbf{i} \in \mathcal{I}}; \mathcal{D}_{N_b}) = \sum_{(\mathbf{i},\mathbf{j}) \in \mathcal{P}} \|k_{\mathbf{i},\mathbf{j}} - p_{\mathbf{i}} \cdot p_{\mathbf{j}}\|^2 \tag{6}$$

*under constraints (5) and where $k_{\mathbf{i},\mathbf{j}}$ is the proportion of same-segment responses for the pair $(\mathbf{i}, \mathbf{j})$ and $\mathcal{P} = \cup_{n=1}^{N_b} \mathcal{P}_n$ is the set of tested pixel pairs (different set of pairs can be tested at each block).*

Under the conditions of Proposition 1, minimizing $\ell$ or $\ell_s$ is equivalent. The proof of Proposition 1 can be found in Appendix A in S1 Text.

## Quantifying the accuracy of the inferred probabilistic segmentation maps

In the following, we refer to the loss $\ell$ defined by Eq (1) as the Binary Cross-Entropy (BCE) and to the loss $\ell_s$ defined by Eq 6 as the Squared Error (SE). In practice, we will always use the SE loss $\ell_s$ as it corresponds to the classical non-linear least-square regression.

We illustrate numerically the theoretical result established by Proposition 1 in Fig 2. Experiments were run with $N_b = 10$ blocks. In practice, we observe that the equivalence of SE and BCE losses holds even if the linear independence condition is not exactly obtained.

First, we compare the SE and BCE numerical optimizations. Both methods find solutions with comparable values of the cost function (light gray lines in top-left and top-middle panels), although convergence is marginally slower for the SE loss function compared to the BCE loss function (note that slower here refers simply to the number of iterations of the numerical optimization, which is distinct from the number of trials $N_t$ collected in an experiment).

As an important additional quantitative comparison between the two methods, we also compute the Mean Absolute Error (MAE) with the ground truth maps. The MAE is defined as the $L^1$ norm of the differences between the K-tuple of the ground truth and reconstructed probability at each pixel, averaged over pixels. Because it is measured on the probabilistic maps, the MAE reflects the accuracy of the estimation of uncertainty. Again we find similar values (light gray lines in bottom-left panel). Lastly, the reconstructed maps are identical

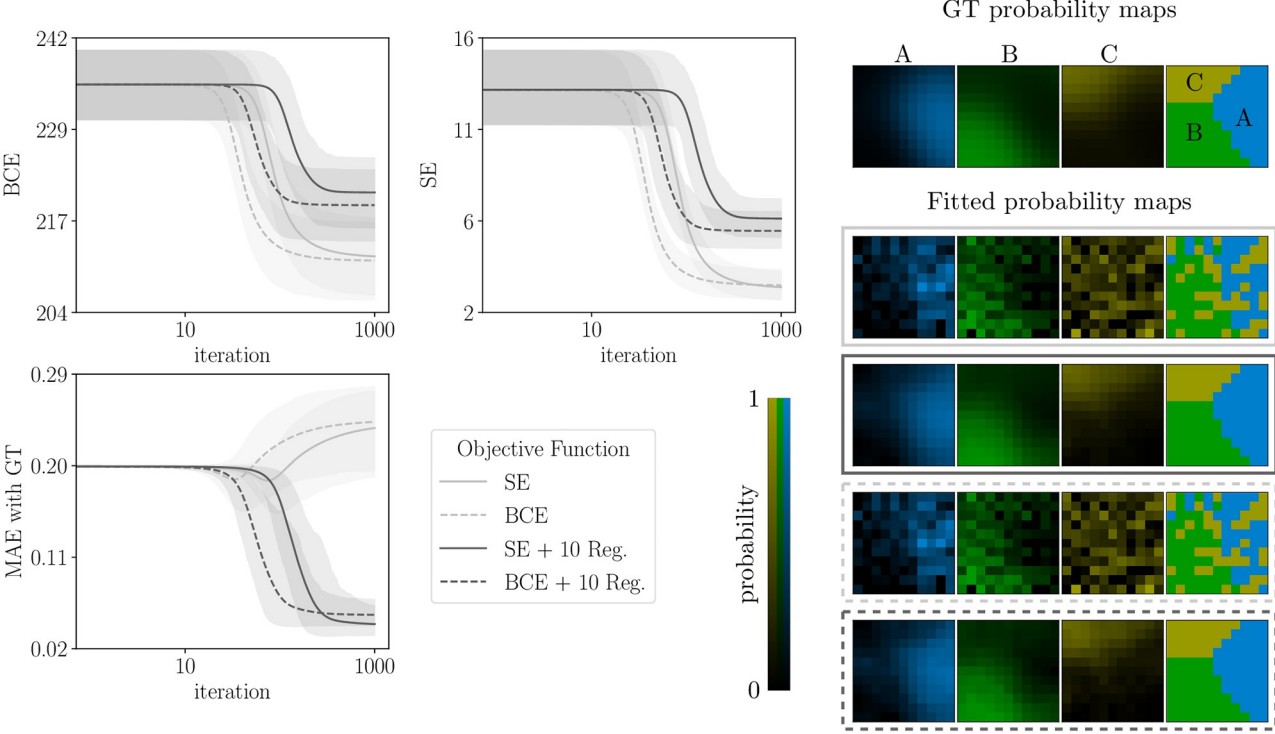

**Fig 2. Equivalence of loss functions and effects of regularization.** Top left: value of the BCE loss when we optimize for BCE (dashed lines) or for SE (continuous lines). Top center: same but for SE loss. Bottom left: value of the reconstruction MAE. In all panels, the shaded areas represent 95% bootstrap error bars over 1000 simulations. Right: ground truth (GT) probabilistic maps and reconstructed probabilistic maps for each objective function indicated in the legend. The mention "10 Reg." means that we use regularization with $\lambda = 10$.

(bottom-right panels). Notice that the MAE increases as the optimization of SE or BCE progresses, confirming the visual impression that both the reconstructed segmentation map and the probability maps are noisy and quite different from the ground truth. In a later section, we show that spatial regularization is an effective solution to this problem.

### Choosing the tested pairs

The probabilistic maps consist of a set of $(K-1)N^2$ unknowns $(p_{\mathbf{i}})_{\mathbf{i} \in \mathcal{I}}$, thus at least $(K-1)N^2$ pairs have to be tested to infer the unknowns (the choice of the number of segments $K$ and the grid size $N$ is further discussed in Appendix C in S1 Text). Proposition 1 narrows the choice of the pairs to be tested: To preserve the relation between the MLE estimates of Bernoulli random variables and the MLE estimate of the probabilistic maps, it is sufficient that for each tested pixel $\mathbf{i}$ the family of probability vectors $(p_{\mathbf{j}})_{\mathbf{j},(\mathbf{i},\mathbf{j}) \in \mathcal{P}}$ is linearly independent.

To gain some intuition about this constraint, consider the deterministic case where the probability vectors $(p_{\mathbf{i}})_{\mathbf{i} \in \mathcal{I}}$ are one-hot vectors (*i.e.* one element equals 1, and all others equal 0). In this case, to preserve the linear independence of the family, a pixel $\mathbf{i}$ must not be tested against more than $K$ other pixels. In addition, it also indicates that the optimal choice of tested pixels is the following: one pixel must be in the same segment as $\mathbf{i}$, the $K-1$ other pixels must belong to every other segments (see Fig 3). As practical guidance for real experiments with natural images, where we do not know the ground-truth segments, a pixel should be tested against a total of $K$ other pixels ensuring that they are sufficiently scattered across the image, given that Gestalt rules suggest nearby pixels are more likely to belong to the same segment than distant pixels. To summarize, we collect enough data to form $KN^2$ equations which is more than the minimal amount that is required ($(K-1)N^2$) but not more to possibly preserve the linear independence of the tested families.

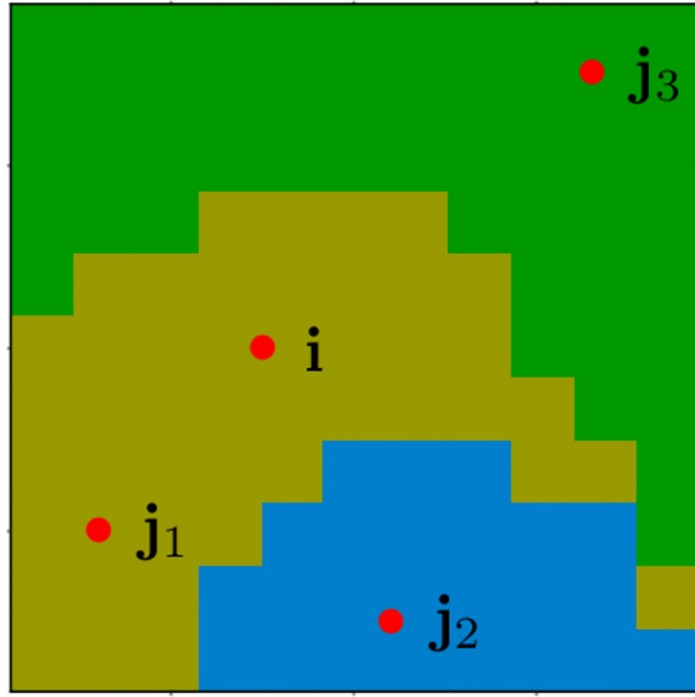

**Fig 3. Optimal choice of tested pairs.** Red dots denote the optimal choice of pixels to be paired with the pixel $\mathbf{i}$, in the case of a deterministic segmentation map.

So far, we have treated the probability vectors at each pixel as free parameters, therefore our approach to reconstruct the probability maps requires optimizing a large number ($(K-1)N^2$) of unknowns. In practice, we find that with limited amounts of data as can be collected in realistic experiments, the reconstructed maps are noisy (illustrated in section *Results*). In the following two sections, we describe two distinct approaches to tackle this problem.

## Spatial regularization

One of the basic Gestalt rules of perceptual segmentation is that spatial proximity encourages grouping [1]. Therefore, although it is still an open question whether human perception uses this rule for grouping and segmentation of complex natural images, we can assume that nearby pixels have a high prior probability of belonging to the same segment when the grid $\mathcal{I}$ is sufficiently fine. We show in the section *Results* that adding such a prior (or regularization) to Eq (6) is a powerful method to reduce noise in the recovered probabilistic maps. The regularized problem writes

$$(\hat{p}_\mathbf{i})_{\mathbf{i} \in \mathcal{I}} = \underset{(p_\mathbf{i})_{\mathbf{i} \in \mathcal{I}}}{\operatorname{argmin}} \sum_{(\mathbf{i},\mathbf{j}) \in \mathcal{P}} \|k_{\mathbf{i},\mathbf{j}} - p_\mathbf{i} \cdot p_\mathbf{j}\|^2 + \lambda \sum_{\mathbf{i} \in \mathcal{I}} \sum_{k=1}^{K} \|p_\mathbf{i}[k] - (G*p)_\mathbf{i}[k]\|^2 \tag{7}$$

where $G * q = \Sigma_\mathbf{j} G_\mathbf{j} q_{\mathbf{i}-\mathbf{j}}$ is the discrete convolution (in practice, edge values of $p[k]$ are repeated to ensure images size consistency), $G$ is a local kernel and $\lambda > 0$. For example, $G$ can be a Gaussian kernel or, as we chose in this paper, a Laplacian kernel. Intuitively, this regularization simply imposes a cost for $p$ being different than the local average over a neighborhood.

In Fig 2 we have illustrated that Proposition 1 still holds, at least approximately, when using regularization; that is, the results for the two loss functions remain numerically equivalent. The effects of regularization are further examined in section Results.

## Parametric models

The model proposed in the previous sections is a parametrization of the probability $p_{\mathbf{i},\mathbf{j}}$ that pixels $\mathbf{i}$ and $\mathbf{j}$ belong to the same segment. We write

$$p_{\mathbf{i},\mathbf{j}}(\theta) = p_\mathbf{i} \cdot p_\mathbf{j} \tag{8}$$

where $\theta = (p_\mathbf{i}, p_\mathbf{j}) \in \mathcal{Q}$ with $\mathcal{Q}$ being the space of parameters. Here $\mathcal{Q} = \Delta_K \times \Delta_K$, the Cartesian product of two $K$-dimensional simplexes. Despite being a parametric model for $p_{\mathbf{i},\mathbf{j}}$, it is the maximally non-parametric model under the assumption of Eq (2). Indeed, we can further consider parametric versions of the underlying class probabilities *i.e.*

$$p_{\mathbf{i},\mathbf{j}}(\theta) = p_\mathbf{i}(\theta) \cdot p_\mathbf{j}(\theta) \tag{9}$$

where $\theta \in \mathcal{Q}$ (with $\mathcal{Q}$ being an arbitrary parameter space). Here, it is unknown if the result stated in Proposition 1 holds under such parametric assumptions. However, we illustrate this approach numerically in section *Results*.

Specifically, we consider feature maps $(x_\mathbf{i})_{\mathbf{i} \in \mathcal{I}}$ associated to the image (for instance, $x_\mathbf{i}$ could be the RGB values of the image pixel $\mathbf{i}$, as in section *Numerical simulation*; or the activation of a bank of visual filters centered at pixel $\mathbf{i}$, as in section *Human participants*). We then define the set of parameters $(\omega, \beta)$ with $\omega \in \mathbb{R}^{K \times D}$ and $\beta \in \mathbb{R}^K$ (where $D$ denotes the feature dimensionality *e.g.* $D = 3$ for RGB features), and consider the following multinomial logistic

model for the class probabilities

$$p_{\mathbf{i}}[k](\omega, \beta) = \frac{\exp(\omega_k \cdot x_{\mathbf{i}} + \beta_k)}{\sum_{l=1}^{K} \exp(\omega_l \cdot x_{\mathbf{i}} + \beta_l)}. \tag{10}$$

The fitting procedure finds the model parameters that best associate the feature map to the empirical mean of the observed samples $(k_{\mathbf{i},\mathbf{j}})_{(\mathbf{i},\mathbf{j}) \in \mathcal{P}}$ (where $k_{\mathbf{i},\mathbf{j}}$ is defined in Proposition 1). See section Discussion for future work on more expressive parametrizations.

## General case

The most general approach is to consider a parameter space $\mathcal{Q}$ and to look for a maximum of the likelihood $\ell$ defined in Eq (1) in the space $\{p_{\mathbf{i},\mathbf{j}}(\theta)\}_{\theta \in \mathcal{Q}}$. Such a problem has been previously explored in the more general case of multinomial distributions but with a single dimensional parameter space *i.e.* $\mathcal{Q} \subset \mathbb{R}$ [57]. With our level of generality it is not known under which conditions the results stated in Proposition 1 hold.

## Implementation and algorithm

We implemented the models described above in Python using PyTorch. In the non-parametric case defined by Eq (8), we use exponentiated gradient descent to perform the inference [58]. The pseudo code implementing this model is described in Algorithm 1. In the parametric case, defined by Eq (10), we use a quasi-Newton gradient descent (PyTorch implementation of the L-BFGS algorithm).

**Algorithm 1**: Inference of probabilistic segmentation maps

```
input : dataset D_{N_b}, number of segments K, learning rate λ_r, stopping
criterion ε
output : probabilistic maps p = (p_i)_{i∈I}
begin
  Initialize u ← 0
  Initialize the probabilistic maps p ← p^(u)
  Initialize the loss values ℓ^(u+1) ← ℓ(p;D_{N_b}) and ℓ^(u) ← ℓ^(u+1) + 1
  while |ℓ^(u+1) - ℓ^(u)| > ε do
    p ← pexp(-λ_r ∇ℓ(p;D_{N_b}))
    p ← p/∑_{k=1}^{K} p[k]
    ℓ^(u) ← ℓ^(u+1)
    ℓ^(u+1) ← ℓ(p;D_{N_b})
    u ← u + 1
  end
end
```

## Simulation details

To validate our methods, we generate synthetic data as follows. Synthetic probabilistic segmentation maps are generated according to the method described in Appendix B in S1 Text. To simulate binary responses $r_{\mathbf{i},\mathbf{j}}^{(n)}$, we first randomly selected a set of pairs $\mathcal{P}$ ensuring that it contains at least once each pixel of the grid. We used the same set of pairs at each block *i.e.* for any block $n \in \{1, \ldots, N_b\}$, $\mathcal{P}_n = \mathcal{P}$. Then, for each pair of pixels $(\mathbf{i}, \mathbf{j}) \in \mathcal{P}$, we sampled $N_b$ Bernoulli variables with parameter $p_{\mathbf{i},\mathbf{j}}$.

In numerical experiments, we re-sampled 1000 times the set of pairs $\mathcal{P}$ in order to show the sampling variability using error bars corresponding to 95% confidence intervals.

## Results

Our goal is to validate our new protocol to measure perceptual segmentation maps, and to demonstrate how it allows us to study uncertainty in human segmentation data. To briefly summarize the procedure detailed above, an experimental session consists of multiple blocks of trials. In each trial, a participant reports if two locations in the image belong to the same segment (Fig 1, bottom-left). We collect binary (same/different) responses at multiple locations, and numerically estimate the underlying probabilistic segmentation maps, *i.e.* the probability that each pixel belongs to any segment, as well as the perceptual segmentation map, *i.e.* the segment with highest probability at each pixel. In Fig 1 (bottom-right) we illustrate these reconstructed maps for one participant with one natural images (additional examples are provided below).

This *Results* section is divided in three parts. We first study the segmentation maps recovered from simulated and real data corresponding to different experimental conditions, offering practical guidance for experimental design. Second, we report the results of a psychophysical experiment on naive human participants with artificial textures, to demonstrate how our method can be applied to study perceptual uncertainty in segmentation. Third, we demonstrate that our approach can also infer the image features used by the participants to perform segmentation, through reconstruction based on parametric models.

### Accurate inference of segmentation maps from synthetic and experimental data

Reconstruction of segmentation maps works perfectly in the absence of uncertainty (*i.e.* each pixel is assigned to a specific segment with probability equal to 1) as illustrated in the top of Fig 1. Conversely, Fig 2 (right panels) shows that when there is uncertainty about the assignment of pixels to segments (which, in the simulated data, translates into variable same/different judgments across blocks), the reconstructed probabilistic maps are less accurate when they are estimated from limited data, as is typical in real experiments. Therefore, we studied in simulations how the accuracy of our approach depends on the level of uncertainty and on the number of blocks $N_b$. Furthermore, the reconstruction algorithm requires specifying a number of segments $K$, therefore we also studied how to deal with experimental data in which $K$ might not be known.

**Robust reconstruction with limited data.** We generated synthetic data with moderate underlying uncertainty, and studied how the accuracy of the inferred maps depends on the dataset size and on the use of regularization (see the section *Spatial regularization*). First, we found that regularization substantially improves the accuracy, *i.e.* it reduces the mean absolute error (MAE) between the ground truth (GT) and inferred maps (Fig 2, bottom left). Importantly, the MAE is measured on the probabilistic maps, therefore it reflects the accuracy of the estimation of uncertainty. This is also appreciable by visual inspection of the reconstructed maps (Fig 2, right panels).

Next, in additional simulations, we studied how the accuracy depends on the number of data points collected. We observed (Fig 4, left) that reconstruction accuracy improved at approximately the same rate with or without regularization, but was 2 to 3 times better on average when using regularization, regardless of dataset size. Upon visual inspection of the maps, regularization afforded near–perfect reconstruction even with only $N_b = 1$ block (*i.e.* corresponding to a single measurement per tested pair; Fig 4 right, example 3), although the MAE shows that accuracy increased quantitatively for larger numbers of blocks, as expected. When using regularization, we observed that the increase in accuracy started leveling off after $N_b = 10$, which can provide a reference for experimental design (for instance, with a grid

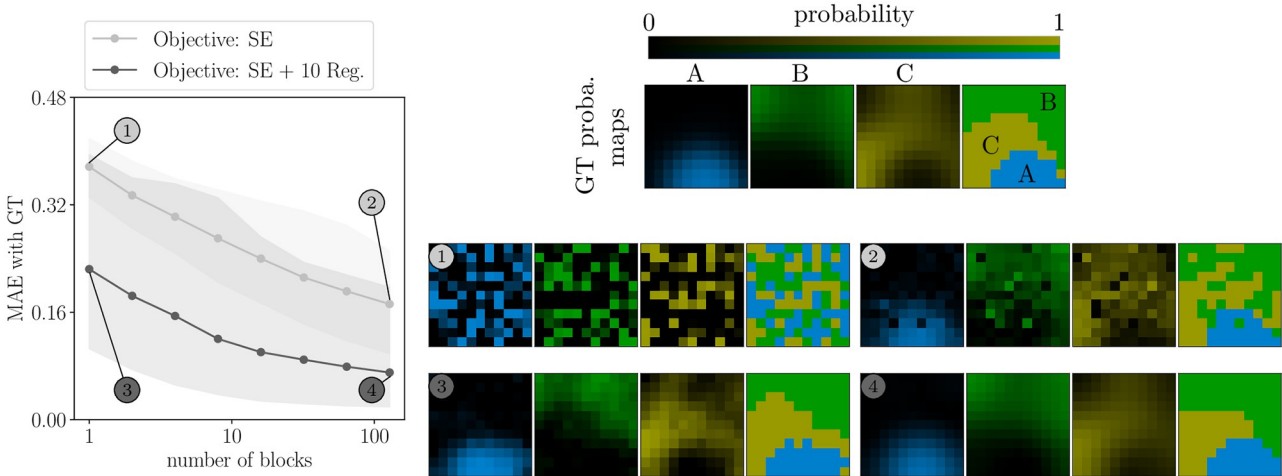

**Fig 4. Accurate inference of segmentation maps from limited data.** Left : the MAE between reconstructed maps and ground truth (GT) as a function of the number of blocks (with and without regularization, light and dark gray respectively). Shaded areas represent 95% bootstrap error bars. Top–Right: ground truth maps. Center–Right: reconstructed maps without regularization from 1 block (left) and 128 blocks (right). Bottom–Right: same as Center–Right but with regularization. The mention "10 Reg." means that we use regularization with λ = 10.

resolution $N = 10$ and $K = 4$ segments, each block lasts approximately 5 minutes, therefore $N_b$ = 10 blocks may be collected in a single session but more blocks might be prohibitive). We also note that this improvement comes at the cost of an increase in variability across simulated experiments (larger error bars with than without regularization, in Fig 4, left), due to the reconstruction bias induced by the regularization.

**Robust reconstruction across levels of uncertainty.** Intuitively, the accuracy of the estimates of uncertainty depends on the estimation of across-block variability, and therefore it could be affected by the ground-truth uncertainty level. Thus, we studied the performance of our reconstruction method for systematic changes in ground-truth uncertainty, with a fixed number of blocks $N_b = 10$. Fig 5 illustrates that the MAE generally increases with uncertainty, because higher ground-truth uncertainty implies noisier observations. When no regularization is used, the MAE rapidly plateaus on average as the uncertainty increases, whereas the MAE variability across experiments decreases. In contrast, when using regularization, the MAE first decreases before increasing strongly for medium levels of uncertainty and then decreasing slightly. The MAE variability is very small for low levels of uncertainty and it is maximal for medium level of uncertainty. Lastly, the reconstruction quality for the two methods is equivalent in the deterministic case, but the reconstructions are 2–5 times better with regularization across all uncertainty levels. These results demonstrate that the regularization enables to robustly capture uncertainty (at least when the uncertainty map has a range of spatial frequency that is similar to the one of the regularization kernel $G$).

**Robust reconstruction with unconstrained number of segments.** So far we have considered the case where we either know the number of segments $K$ in the ground-truth synthetic data, or, in the real experiments, we ask the participants to partition the image using a specific value of $K$. However, in some variations of our experiment, we would like to measure segmentation maps without specifying the number of segments. For instance, this is relevant for natural images where there is no obvious ground truth, or for artificial images with high uncertainty, where the perceived number of segments could be an additional source of variability.

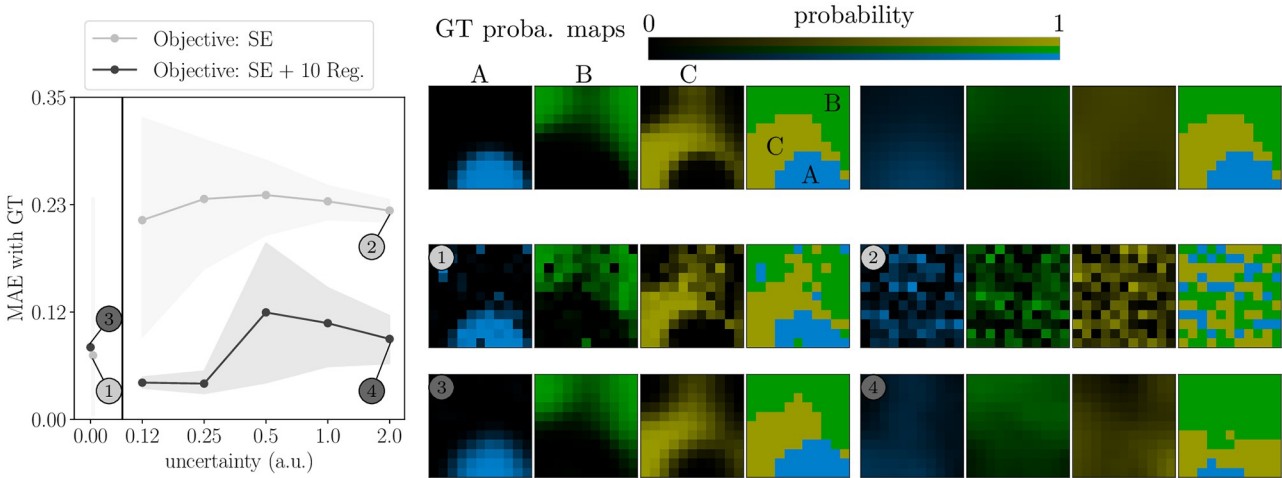

**Fig 5. Accurate inference of segmentation maps from variable data.** Left: the MAE between reconstructed maps and ground truth (GT) as a function of the uncertainty (with and without regularization, light and dark gray respectively). Shaded areas represent 95% bootstrap error bars. Top–Right: ground truth maps. Center–Right: reconstructed maps without regularization from low (left) and high (right) uncertainty. Bottom–Right: same as Center-Right but with regularization. The mention "10 Reg." means that we use regularization with $\lambda = 10$.

Therefore, we first verified in simulations that when uncertainty is moderate and when using regularization, the value of $K$ for the reconstruction can be determined with a straightforward approach: We generated data using $K = 5$ segments, and reconstructed the maps using $K = 3, 4, 5, 6$ and 7. In the reconstructions using $K = 6$ and 7, the superfluous probabilistic maps were automatically set to zero. Therefore the correct $K$ can be inferred from the reconstructed maps, as the maximum value of $K$ that produces no empty maps (see Appendix C in S1 Text).

Next, we conducted experiments with human participants segmenting natural images. Participants were not instructed about the number of segments, and instead were informed that the level of detail in segmenting the images was up to them. The results are presented in Fig 6. We performed the reconstruction assuming $K = 5$ segments, but we recovered only 3 segments in most images, except for the 6th and 8th images for which we recovered only 2. According to the simulations described above, those numbers are likely to reflect the true number of segments used by the participants on average in the aggregated data.

Interestingly, our approach also revealed that regions of high perceptual uncertainty can be captured in the probabilistic maps, even when those regions do not account for a segment in the deterministic segmentation maps. For instance, in the fifth probabilistic map in image 8, the dry grass on the top is sometimes grouped separately from the ground, but most often the two are grouped together in the segment corresponding to the second probabilistic map. Regions of high perceptual uncertainty are also evident in other images, such as in image 7, where the branches are only partially occluding the background sky, so the pixels around those are sometimes grouped together with the bottom branches and sometimes with the background sky. One caveat is that here we reconstructed the maps from the aggregate data across participants (see section Materials and methods, Experimental Participants), therefore these observations may reflect variability across individuals, and we did not assess per-participant uncertainty. In the next section, we examine more closely how our approach can be used to study the uncertainty of perceptual segmentation at both individual and aggregate levels.

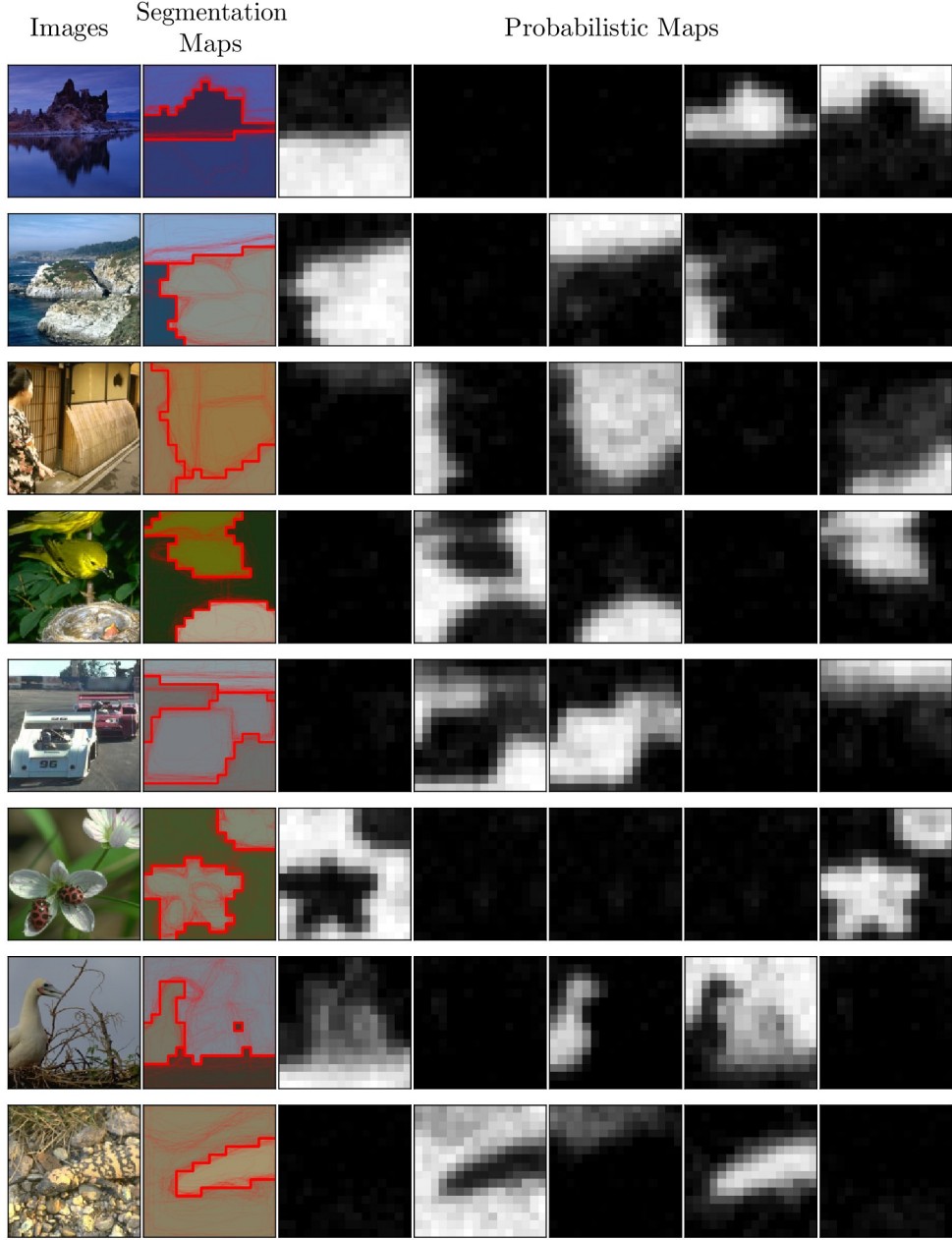

**Fig 6. Human Segmentation of Natural Images.** From left to right: the original images, the corresponding segmentation maps, and the five corresponding probabilistic maps. Maps were reconstructed with regularization ($\lambda$ = 5).

## Measured uncertainty in human participants correlates with image uncertainty

To demonstrate the use of our method to study human visual segmentation, we conducted a pilot study online in which we manipulated the segmentation-related uncertainty in artificial images (see Appendix B in S1 Text). Note that our goal here is to reconstruct and analyze the probabilistic maps of each individual participant, not those reconstructed from aggregate data

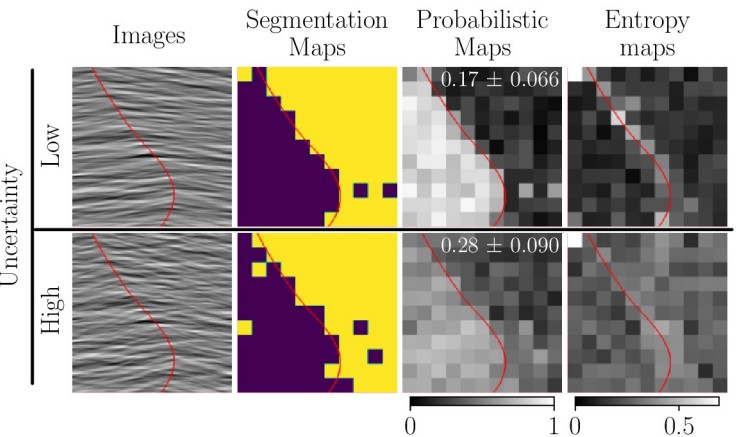

**Fig 7. Variability in human segmentation reflects image uncertainty.** From left to right: tested images, segmentation maps, probabilistic maps of the left region and entropy maps corresponding to the reconstructed probabilistic maps *i.e.* $p_i[1] \log (p_i[1]) + p_i[2] \log (p_i[2])$ (average entropy ± 3 standard errors is indicated by the text in white). Top: low uncertainty case (texture orientation distributions are weakly overlapping). Bottom: high uncertainty case (texture orientation distributions are strongly overlapping). In all panels, the red line represents the ground truth boundary between the two segments (shown only for visualization purposes, not in the real experiments). Maps are reconstructed without regularization ($\lambda = 0$).

as in the previous section. We analyzed data from 15 participants in the low-uncertainty and in the high-uncertainty conditions. To not bias our analysis towards reconstructing smooth probabilistic maps we have not used regularization *i.e.* $\lambda = 0$. We observe that the segmentation maps (Fig 7, second column) are very similar between conditions, except for a few, noisier pixels in the high-uncertainty condition. However, we find that the measured uncertainty of the inferred probabilistic maps (*i.e.* the total entropy of the maps; Fig 7, numbers in the third column) is significantly larger for images with higher segmentation-related uncertainty (Cohen's $d = 1.003$, Welch's t-test with $t = 2.746$ and $p = 0.0109$). Furthermore, the entropy maps reveal a spatial structure that suggests the measured variability does not simply reflect noise: when uncertainty is low, human-uncertainty is localized around the edge between textures, whereas when image uncertainty is high, human uncertainty is more uniformly spread across the entire image (see also individual entropy maps in Appendix E in S1 Text). These results highlight the importance of measuring the variability and uncertainty of human segmentation, and they are consistent with the hypothesis that perceptual processes underlying segmentation include a correct representation of uncertainty [37, 59]. As we show in the next section, these measurements of variability also allow us to compare models of perceptual uncertainty and reveal the image features that participants use to perform segmentation.

### Fitting parametric models to infer the image features used for segmentation

We have shown in section *Parametric models* that the hypothesis of the existence of underlying probabilistic segmentation maps can be strengthened by the additional assumption that they are parametric probabilistic maps, which depend on some features of the image (Eqs (9) and (10)). In other words, with this approach it is possible to use the measured data to estimate the parameters of any hypothesized relation between features of the image and the probability that each pixel belongs to any segment, *i.e.* the parameters of a segmentation model or algorithm. The motivation for fitting such parametric models is twofold: (i) it will allow quantitative

model comparison and hypothesis testing of perceptual segmentation theories and, (ii) it offers the opportunity of finding models that are more data-efficient than the non-parametric model.

**Numerical simulation.** We first validated the parametric approach in Fig 8. We generated images whose features are the color values (a 3–dimensional vector *i.e. D* = 3) of each pixel, and these color features are sampled from a generative model with different parameters for each segment (see Appendix B in S1 Text for details). We use a high resolution ($N$ = 48) in this simulation to provide more samples when training the model and therefore identify the clusters more accurately. We then generated $N_b$ = 10 blocks of simulated data, and applied our inference algorithms.

The parametric model correctly recovers the probabilistic maps up to some noise matching the sampling noise of the image color features (in Fig 8, compare the features in the top–right and the reconstructed probabilistic maps in the bottom–left). Importantly, the parametric model also properly characterizes the features associated to each segment by a single 3–dimensional vector (see the bottom–right scatter-plot in Fig 8).

**Human participants.** Having validated the parametric approach, we further illustrate its power by applying it to our human data. To this purpose, we use a reparametrized version of the model defined by Eqs 9 and 10 with $K$ = 2. Specifically, for $k \in \{1, 2\}$,

$$\beta_k = 0 \ \text{and} \ \omega_k = -\frac{1}{\sigma_k^2} \ \text{with} \ \sigma_k \in \mathbb{R}^D$$

where the inverse is taken component wise. Such a paramatrization allows to interpret $\sigma_k^2$ as the average feature energy. Because the textures used in the experiment are generated as superpositions of wavelets (Appendix B in S1 Text), we defined for each pixel **i** the feature $x_\mathbf{i} \in \mathbb{R}^D$ as the vector of average wavelet energy, with $D$ = 36 orientation bands. The average is calculated over all wavelet scales and pixels in a small square, partitioning the stimulus in a grid of size $N$ (matching the experimental grid $\mathcal{I}$). As $K$ = 2, Eq (10) can be simplified revealing that only the vector difference

$$\omega_1 - \omega_2 = \frac{\sigma_1^2 - \sigma_2^2}{\sigma_1^2 \sigma_2^2}$$

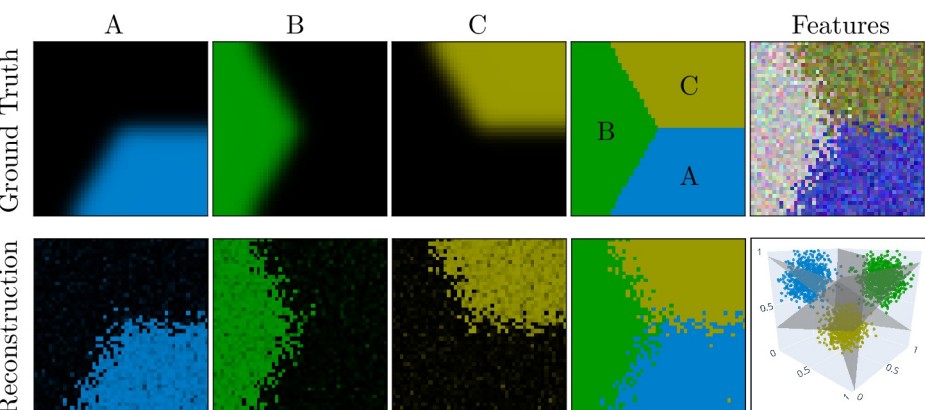

**Fig 8. Validation of the parametric approach.** Reconstruction using a parametric model for the class probabilities (Eq (10)). Reconstruction was achieved minimizing the SE with regularization ($\lambda$ = 1) Left: probabilistic maps and segmentation maps. Right: features displayed as an image and as 3d points in the RGB cube with the planes separating each pair of segments.

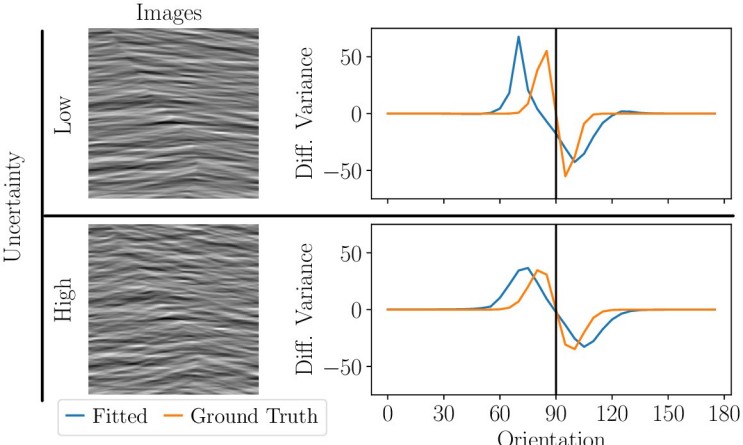

**Fig 9. Uncertainty modulates the perceptual mapping between features and segments.** Left: tested images (same images and data as Fig 7). Right: differential variance (or weight vector, see main text) best relating oriented wavelet features to human responses. Top: low uncertainty case (texture orientation distributions are weakly overlapping). Bottom: high uncertainty case (texture orientation distributions are strongly overlapping). Maps are reconstructed without regularization ($\lambda = 0$).

is relevant for the fitting. Again, to not bias our analysis towards reconstructing smooth probabilistic maps we have not used regularization *i.e.* $\lambda = 0$. Yet, note that the parametric model at use provides another type of regularization (see section Materials and methods). Results are shown in Fig 9 where we compare the fitted vector of differential variances $\sigma_1^2 - \sigma_2^2$ to the ground truth (*i.e.* corresponding to the energy of the image in each of the 36 orientation bands). Qualitatively, the participants correctly attributed more weight to the relevant orientations around 90 degrees, although we also observed a small bias away from 90 degrees compared to the ground truth (compare orange lines *i.e.* the true distribution in the textures, versus blue lines from the fitted parameters). This bias could reflect that this is where the orientation energy distributions of the two textures are least overlapping. In addition, the widths of the bumps are larger for the high uncertainty condition than for the low uncertainty condition, consistent with the ground truth. This indicates that participants integrated information over a broader range of orientations when image segmentation uncertainty was larger, further supporting the hypothesis that uncertainty plays an important role in perceptual segmentation.

## Discussion

We have introduced a well-controlled and standardized protocol to measure probabilistic visual segmentation maps. The protocol collects multiple same-different judgments on the same image, and performs model–based reconstruction of probabilistic segmentation maps, *i.e.* a decomposition of the image into its visual objects together with the probability that any pixel belongs to any object. First, we have demonstrated our approach with both simulated experiments using synthetic images, and experiments with human participants segmenting natural images. We have found that appropriate regularization is necessary to obtain robust reconstructions of segmentation maps and their uncertainty, with realistic amounts of data and across a range of experimental conditions (Figs 4, 5 and 6). Second, we have shown that the reconstruction can be either non–parametric, or based on any parametric segmentation algorithm, therefore our protocol enables fitting any such algorithm to the data. We have illustrated this with parametric models where the probabilistic segmentation maps capture the

statistical regularity of object features (colors or orientation content), and we have shown with both synthetic (Fig 8) and real experiments (Fig 9) that the same/different data are sufficient to accurately estimate those regularities. Lastly, our results revealed that measured variability in human perception correlates with segmentation-related uncertainty qualitatively (Fig 6) and quantitatively (Fig 7), and that participants correctly weigh relevant image features differently depending on uncertainty (Fig 9). Therefore, our work indicates that measuring and modeling segmentation uncertainty will be important to test theories of perceptual segmentation and to better quantify the performance of segmentation algorithms.

Our protocol closely integrates two key innovations to substantially improve over existing approaches to study segmentation. First, it relies on repeated trials that accumulate same/different perceptual decisions to a single pair of points on an image. Same/different judgments is a classical paradigm in visual psychophysics [38], yet it had not been used before to measure full segmentation maps. Thanks to this approach, our method addresses the three main shortcomings of existing segmentation databases used so far in computer vision that are typically based on manual tracing of contours [29–33]. The first shortcoming is that manual tracing introduces biases and variations unrelated to perceptual processing. The manual tracing task can bias a participants to draw smoother contours than perceived, because that requires less effort, and can add variability across individuals due to uncontrolled variation in effort level. In our task, the effort required to *report* a perceptual judgment does not depend on the smoothness of the contours. Importantly, the effort to *reach* that perceptual judgment certainly depends on the visual features (including contour smoothness), and our method measures potential behavioral correlates of that effort, *i.e.* reaction times and across-trial variability. The second shortcoming of existing databases is the lack of control and measurement of timing, which introduces another factor of variation unrelated to visual processing. Our protocol precisely controls the presentation time: the total presentation time of an image throughout the session is identical across participants and across images, and the per-trial presentation time of the image with the cues is identical across trials, across images, and across participants. The third shortcoming is the lack of measurements of perceptual variability for each individual participant. With our method, repeated measurements of the same pairs allow us to quantify variability, and the number of repetitions can be chosen based on a tradeoff between the resolution on the measurement of uncertainty and the spatial resolution, given a desired duration of the experiment. Importantly, for applications in which variability is not of interest, we have shown that the deterministic segmentation map can be reconstructed from measurements of a single trial.

Our second key innovation is to use model–based reconstruction of segmentation maps. Inference of those segmentation maps can be achieved in practice by either minimizing the least square errors or by the classical maximum likelihood estimation of the probability of a Bernoulli random variable. We have shown that the two approaches are equivalent under mild conditions. Our model–based reconstruction has broad potential implications both for vision research and for artificial intelligence. To perform the reconstruction, one has to specify a parametric model of the segmentation map (either deterministic or probabilistic), namely a model that computes the segmentation map given an image and a set of parameters that relate image pixels or features to image segments. Given one such segmentation model or algorithm, the reconstruction works by finding the parameters that produce the segmentation map most consistent with the collection of same/different judgments. This opens up two broad directions for future applications. The first one is to collect enough data on individual participants to constrain models that implement specific hypotheses about visual segmentation, and compare them quantitatively using the same data and cost function. The second direction is to use our protocol for massive online data collection to create the first dataset of purely perceptual

segmentation maps, along with clearly defined benchmark metrics. Creating benchmarks based on intra-subject variability would be particularly interesting and novel. The `vseg` python package we have provided (https://vseg.gitlab.io/vseg/) includes code that automates remote data collection, and it allows to seamlessly plug in any segmentation algorithm, thus facilitating both applications described above.

Our experiments relating segmentation uncertainty to measured human variability (Fig 7) offer a concrete demonstration of the first direction. Uncertainty is a central concept in theories of perception in general [36], and segmentation in particular is thought to require probabilistic inference [37] because image pixels often cannot be assigned to a specific object with full certainty. The experiments of Figs 7 and 9 demonstrate how our protocol could be used to test this hypothesis. Specifically, we have generated composite texture images from a simple probabilistic generative model, *i.e.* a Gaussian distribution over orientation, with different mean (center orientation) in each segment, and we have manipulated the ground-truth uncertainty by changing the similarity of the parameters of the texture in each segment (*i.e.* their orientation bandwidth). We have found that the variability of the human segmentation maps increases for images with higher uncertainty (center of Fig 7), that it is concentrated near areas of higher uncertainty (the boundary between textures; center and top–right panel of Fig 7), and that the fitted parameters, *i.e.* the weights placed on each orientation band, reflect the ground-truth uncertainty (*i.e.* integration over a broader range of orientations when uncertainty is higher; Fig 9). However, we emphasize that this was not meant as an exhaustive test of the hypothesis, only as an illustration of how our protocol could be used to test it. That will require collecting datasets with more trials and conditions to better constrain the parameters for individual participants, and comparing the reconstruction model used here (based on probabilistic inference) against alternative, including popular models based on feature discrimination [59].

There are several other uses that our method and its future extensions will enable. First, as explained above, our method focuses on perceptual factors and reduces the effect of other confounders of datasets created with manual tracing. Therefore, it can be used to improve understanding of the potential biases (or lack thereof) in measuring segmentation using more traditional methods. Second, this represents an opportunity to compare contour–based segmentation (as in tracing tasks, where the participants indicate whether a pixel is a boundary of a given object, rather than the segment label of each pixel) and region based segmentation (as in our method, where the task is to compare the image regions around the two cues). Third, different from tracing tasks, our method employs a trial-based design, with precise control of cues and stimulus onset/offset. This would facilitate analyzing and interpreting concurrent recordings of brain activity, e.g. with EEG, MEG or fMRI. Furthermore, because the basic unit of our task is a simple discrimination, it may be possible to train animal models on variants of our task and thus study the neural bases of natural image segmentation with possibly invasive recordings and perturbations.

Although we have extensively validated the protocol with synthetic experiments, and demonstrated its applicability in real experiments, the novelty of our method leaves ample room for improvement. First, because of the minimum requirement on the number of trials, the time needed to collect enough data for one image scales linearly with the number of segments and quadratically with the spatial resolution. This makes it impractical to collect high resolution maps with individual participants, due to the long duration of the experimental session. One attractive solution is to use model–based reconstruction, which can drastically reduce the minimum number of trials, but other options should be explored. A different avenue is indicated by our demonstration that segmentation maps can be successfully reconstructed from aggregate data across participants: high resolution maps could be obtained by collecting only a few trials from each of a very large number of participants, which is feasible with

crowdsourcing. Second, in all cases tested, we have found that the method is more robust when using Laplacian regularization than no regularization. However, there is no clear principle to select the regularization parameters ($\lambda$, $G$), and more generally it is possible that other regularization schemes or priors could improve performance. Third, our Proposition 1 suggests a strategy to select the pairs of image locations used for the measurements, but there may be better choices in other settings. Fourth, it will be important to develop extensions that avoid using an underlying grid of tested locations, that accommodate variable resolution to focus on image areas that are most informative (*e.g.* for comparing specific hypotheses or algorithms), and that do not have a strict constraint on the minimum number of pairs (see also the robustness of the proposed algorithm to increases in resolution in Appendix D in S1 Text). Lastly, we have introduced parametric models of the segmentation maps (*e.g.* Eq (10)) and have emphasized that they allow for relating the segmentation maps to image features. Such a parametric approach includes deep neural networks parametrized by their weights. However, deep neural networks will only be trainable once sufficient amount of data is available.

## Supporting information

**S1 Text.** Appendix A. Proof of Proposition 1. Appendix B. Stimulus generation. Appendix C. Large or unknown number of segments. Appendix D. Resolution of the segmentation maps. Appendix E. Individual entropy maps. **Fig A. Large Number of Segments.** To test the feasibility of the reconstruction for a large number of segments, we generated an artificial segmentation map with $K = 9$ segments and $N = 25$. The reconstruction obtained from measuring a single repetition of the minimal set of pairs, remains accurate when using spatial regularization. Top: ground truth. Center: no regularization. Bottom: Laplacian regularization. **Fig B. Unknown Number of Segments.** Reconstruction using different values of $K$ with regularization. Top: ground truth. Then, from top to bottom, reconstruction with $K = 3, 4, 5, 6$ and $7$. If the true $K$ is unknown, it can be correctly inferred from the reconstructed maps, as the maximum value of $K$ that produces no empty maps. **Fig C. Resolution.** Effect of increases in resolutions over the reconstruction of probabilistic segmentation maps. Top-left: ground truth maps. Top-right: reconstruction without regularization. Bottom-right: reconstruction with Laplacian regularization. MAE between the reconstructed maps and ground truth is indicated on top of each collection of maps. Bottom-left : MAE between the reconstructed maps and ground truth as a function of the resolution. Shaded areas represent 95% bootstrap error bars. **Fig D. Wider Kernel Regularization.** Effect of the kernel width used for the regularization. This must be compared to Fig C bottom-right. **Fig E. Individual entropy maps.** Top-left: the 15 participants in the low uncertainty condition. Bottom-left: the 15 participants in the high uncertainty condition. The contour drawn in red is drawn by the participant. Bottom-right: distribution of the contour f-scores of the participants. **Table A. Summary of the stimulus parameters.** Parameters of the stimuli used in the experiments.
(PDF)

**S1 Video. Online experiment example.** Video illustrating the sequence of screens a participant has seen before starting the experiment.
(MKV)

## Author Contributions

**Conceptualization:** Jonathan Vacher, Pascal Mamassian, Ruben Coen-Cagli.

**Data curation:** Jonathan Vacher.

**Formal analysis:** Jonathan Vacher.

**Funding acquisition:** Pascal Mamassian, Ruben Coen-Cagli.

**Investigation:** Jonathan Vacher.

**Methodology:** Jonathan Vacher, Pascal Mamassian.

**Project administration:** Pascal Mamassian, Ruben Coen-Cagli.

**Resources:** Pascal Mamassian, Ruben Coen-Cagli.

**Software:** Jonathan Vacher.

**Supervision:** Pascal Mamassian, Ruben Coen-Cagli.

**Validation:** Claire Launay.

**Visualization:** Jonathan Vacher, Claire Launay, Pascal Mamassian, Ruben Coen-Cagli.

**Writing – original draft:** Jonathan Vacher.

**Writing – review & editing:** Jonathan Vacher, Claire Launay, Pascal Mamassian, Ruben Coen-Cagli.

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
