## [Decision Letter · Decision Letter 0]

11 Apr 2023

Dear Prof. Vacher,

Thank you very much for submitting your manuscript "Measuring uncertainty in human visual segmentation" for consideration at PLOS Computational Biology.

As with all papers reviewed by the journal, your manuscript was reviewed by members of the editorial board and by several independent reviewers. In light of the reviews (below this email), we would like to invite the resubmission of a significantly-revised version that takes into account the reviewers' comments.

Please pay specific attention to the reviewers concerns over the mathematical notation and the phrasing of the paper with respect to how it complements the existing literature.

We cannot make any decision about publication until we have seen the revised manuscript and your response to the reviewers' comments. Your revised manuscript is also likely to be sent to reviewers for further evaluation.

Sincerely,

Emma Claire Robinson

Academic Editor

PLOS Computational Biology

Daniele Marinazzo

Section Editor

PLOS Computational Biology

Reviewer's Responses to Questions

**Comments to the Authors:**

Reviewer #1: Review of "Measuring uncertainty in human visual segmentation"

Reviewer: Michael Landy

This paper describes a technique for determining perceived image segmentation on a pixel-by-pixel (or rather, grid-block-by-grid-block) basis using choice data. It is a substantial contribution and should be published. I have some substantive comments below, but they are all easily fixable. My main kvetch is that the math is needlessly obscure and could use some clarity to make it easier to follow.

Specifics, mostly by line number:

* OK, this one isn't by line number. Decades ago, my friend Pete Mowforth (once a co-founder of an image-processing group in Glasgow called the Turing Institute) sent out a natural image to a bunch of well-known researchers in image processing and computer vision (think: people on the order of Azriel Rosenfeld) and asked them to mark where the edges were (not identical to asking for image segmentation, but close). The results were wildly variable across people, with some indicating any color or luminance boundary, and others indicating object boundaries only, including completion of boundaries through locations where no actual luminance/color boundary was in the image, etc. The point is that "segmentation" is necessarily multi-scale and ill-defined. Yet, the main work here involves combining data across participants and there really isn't enough discussion about the importance and meaning of inter-participant differences in strategy in this task.

* 111-116: This is about the first time (well, slightly above) that we see the word "grid" and the value of N is given even though N has not yet been defined. So, clarify that you'll be asking for judgments on a square grid with a resolution coarser than the pixel resolution in the images. Clarify somewhere where the red dot is placed within the grid square. Clarify why N_t = K N^2. At this point, I'd think that there are N^2 grid squares and you are testing arbitrary pairs, so that the number of points tested should be on the order of N^2-choose-2 (i.e., on the order of N^4) and that the value of K isn't relevant. So, this definition of N_t comes out of the blue here. And why does the number tested (per block) drop to (K-1)N^2 5 lines later?

* 173: The "grid" is defined to be "script-I" on 158 and yet here you state that the pairs are drawn from script-I^2. And, you now act like script-I isn't the grid (the set of grid squares), but rather the pixels inside those grid squares that will be tested. Unclear.

* Eq. 1: This is the most needlessly obscure and, frankly, incorrect piece of the math. Anyone who has coded a psychometric function fit would know that the likelihood of a single trial is either p or 1-p, depending on the response. Not everyone has seen the shorthand of coding the likelihood as p^r*(1-p)^(1-r), and this shorthand depends on coding r as 0 or 1, which you never state overtly. Finally, stating the log of this as the sum of KL distance and cross entropy is both unnecessary and incorrect. "r" is not a probability, it's an actual response. Its value is zero or one, so you can't take one or the other log in the definition of KL. Sure, if you act like you can take this log and write out the sum of KL and H, terms cancel and you end up with the correct expression. But why do this?

* 185: Here's my main confusion with the entire paper: I'm not sure what it means that the segment assignments are "independent". I think what you mean, and this should be clarified, is that when an observer, whom we are modeling with these k-tuples of probabilities per grid square, is asked about a pair, the observer gets an assignment for each element of the pair, drawn from the multinomial indicated by the grid squares k-tuple, and the draws for the two grid squares are independent. That's what is required for Eq. 2 to be true. I got distracted by the idea that where the probability assignments came from was independent or some such obscure complexity. So, clarify that what you mean is actually about your model of the observer.

* Throughout, there's the usual issue with k-means and other clustering algorithms that the cluster "names" can be permuted and fit just as well. Here, the assignments of segments to the K labels is arbitrary and so multiple runs will likely land on different permutations but be otherwise equivalent. Worth a comment/clarification?

* 194: l_0 is MINIMIZED when...

* 199-200: Here is a good example of why I'm still confused about what you mean by "independent". You say the family is independent, but that sounds like you are treating the p's as random variables, rather than treating the grid squares' assignments (based on the p's) as random multinomial variables. This phrasing leads me astray.

* Proposition 1: Again the notation is needlessly obscure, especially the use of the indicator function. Somewhere you should say in plain English what these things are (the k_{i,j}'s are just the proportion of same-segment responses for pair i,j and the formula is computing squared error relative to the predicted probabilities of those responses; N_{i,j} is just the number of times i,j was tested and from your description earlier, that shouldn't depend on the pair i,j, I'd think.

* A minor harumph: I generally disklike it when a paper invents a bunch of non-standard acronyms and expects the reader to remember them all. The is worse when those acronyms might already have a meaning in the reader's head. I tend to read SE as standard error, MAE as motion after-effect, BCE as "before the common era" (politically correct for BC) and GT as "greater than". Bleccchhhh. BCE is particularly bad, since its meaning is an obscurity to begin with rather than simply referring to log likelihood. MAE is also confusing since it's defined on pairs of 3-tuples, so how do you turn that into scalar error?

* Fig. 2, lower-left panel: Do you ever remark on the fact that the unregularized MAEs get worse as the iterations proceed? Is MAE the right way to think about error anyway, since it depends only on the two outliers?

* 247: This is yet again an example that seems to mean a different kind of independence than I thought you meant. So, clearly I remain confused about this.

* Eq. 7 and the short text afterward: Again, this is just stated in math and not in English. If G is a Gaussian, then you are imposing a cost for p being different than the local average over a neighborhood, which is pretty straightforward. I'm not sure what a Laplacian plus a constant (is \\delta a constant???) buys you, but I would imagine that i's p-value should be compared to a local average over a neighborhood that omits pixel i. Clarify.

* "Classes of a pixel pair are independent": Every time the wording about independence comes up, I only get more confused. Most of these wordings sound like the pixel K-tuples "p" are random vectors and thus independence for different pixels means they are drawn independently from whatever distribution these K-tuples come from. I doubt you mean that, but I'm lost.

* 280 through Eq. 10: Again this is needlessly obscure and could be made more readable.

* 380: ... robustly capture uncertainy if uncertainty is as spatially lowpass as your regularizer/blur kernel.

* Around Figure 5: I got the impression that you are getting pretty segmentations with only 2 or 3 segments because of pooling over observers. If some observers have fewer and some have more segments, the pooling will end up being well fit by the lowest number of segments on which the subjects tend to agree. I might be wrong, but I think your result of so few segments likely is a result of this. For example in Figure 6, in the first row I'd imagine some observers might have made the reflection a segment, or in the 2nd the 3 rocks as separate segments, or in the 5th row they may have made separate segments for the two cars, but combining subjects ended up merging these separate "objects".

* 432-433: I don't see how these results imply the observers have an estimate of their uncertainty (although I've done my best, in other contexts, to prove just that!).

* S1, first display equation: Shouldn't the sum go from 1 to N_b, not N_t?

* S2, the first display equation and the equations in the line following: I'm lost here as well. Why does the "Gaussian"s exponent have a term ||i||^2??? The pixel index??? I couldn't clearly parse any of this and the term "Gaussian random field" doesn't help me figure out what the equations are doing.

Reviewer #2: The authors present a novel method for extracting segmentation masks from time-limited human same-different forced choice trials. They show that they are able to reconstruct segmentation maps from such trials reliably using both human psychophysics and simulation, and they discuss in detail the specific experimental setups and potential methods for reconstructing segmentation masks from human trials.

Overall, I believe the work is scientifically solid but I have major concerns about the phrasing of the work, particularly in the introduction, as well as some concern about its usefulness and whether the underlying task structure is any more quantitative than extant segmentation methods.

That being said, as the authors point out in the discussion of the paper, there are foreseeable ways to improve the general procedure both conceptually and practically in future work. The current work, to the best of my knowledge, makes the first serious attempt at computing segmentation masks from well-controlled psychophysical experiments and I believe this work could serve as an important starting point for other work to build upon, could lay the groundwork for evaluating the validity of segmentation masks in simple stimuli, as well as provide insight into the human visual system in small-scale experiments.

Major comments

1. The authors state that there are at least three shortcomings with existing human segmentation databases. While I agree these are all potential problems of extant segmentation databases, I am not convinced that the present method convincingly fixes any of them. Below I list the statements and my responses to them:

a. Existing databases rely on manual tracing of contours, introducing interactions between perceptual processes, motor planning and execution, and motor noise.

It seems difficult to me to be convinced that the interaction between perception and motor planning/execution, let alone motor noise, play an important role in current segmentation datasets. This can be quantified: in the Cityscapes segmentation challenge (Cordts et al., 2016), the authors assessed the quality of their labeling by having different annotators label the same images. The images had 98% pixel-level agreement between annotators. This level of agreement suggests a hard upper bound on motor noise, planning and execution that is very low (I suspect much lower than the error imposed by the very low resolution of the perceptually produced segmentation masks in this work).

b. Existing databases have no constraints or measurements of timing.

It is true that in most/all segmentation databases there are no controls of timing. This is an issue, but I wonder to what extent this aspect is meaningfully controlled for in the current paper’s experiments. Participants are shown the same stimulus in the same location for up to an hour – the demo file attached to the paper leaves me unconvinced that many of the desiderata of time-controlled experiments (confounds of perception, decision making, attention, etc.) are fulfilled.

That being said, I do foresee how it could be if the number of trials needed per image was lowered by a meaningful margin. If this was possible, one could randomly show any of a number of images to a participant, potentially mitigating at least some of these top-down effects that may be present in the current paradigm.

c. There are few cases of inter-subject variability on an image-to-image basis in existing databases.

That there are few cases of inter-subject variability on an image-to-image basis in extant databases is often true, but similarly the masks are often validated across subjects in smaller samples (as in the case of Cityscapes, with little variation in masks). In addition, if I have understood correctly, this is not an inherent advantage of the proposed method, but a general comment on collecting segmentation masks. I do not see why more classic methods (like drawing) could not measure inter-subject variability at scale if it was deemed an important experimental question for the purpose. That most current benchmarks do not do this seems unrelated to the proposed methodology given that an actual benchmark of human segmentation masks is not put forth.

A closer justification of the aforementioned issues in the paper is needed, as the paper’s justification for the usefulness of its method is not currently appropriately justified.

2. Following up on that, given that the paper claims problems with existing methods of segmentation map estimation, and claims to solve these problems, it seems important to me to have at least a small control experiment where this is quantified. How different are the segmentation maps in a free-drawing experiment using e.g. LabelMe (Russell et al., 2007) and the current method? Can between-subject uncertainty be estimated using traditional methods (this could potentially be tested by measuring the per-pixel variance across subjects)? I am not sure why the proposed method is superior in this respect given that it does not seem that within-subject variance is measured in e.g. the section starting at line 415.

This need not be a separate large-scale experiment – I would be satisfied with seeing a small pool of participants complete a free-drawing task on the same images they segmented using the proposed method, downscaled to and evaluated at the same resolution as the proposed method. It would be important to me to know whether there are crucial perceptual factors or biases that this experiment can capture with its more controlled design than a typical free drawing task.

3. I am concerned with the number of trials needed to estimate a segmentation mask, as well as with the resolution of the segmentation mask. According to the paper it takes a single participant approximately 50 minutes to measure the segmentation mask to a single stimulus at 11x11 resolution and 2 objects. I use the comparison of the CityScapes dataset again as it illustrates the point clearly: the CityScapes dataset reportedly takes approximately 90 minutes per image to segment, has a resolution of 1280x720 with up to 19 unique semantic classes, with potentially many instances of each. To collect high-quality segmentation masks with even a fraction of that resolution using the method presented in the paper, one would have to perform prohibitively expensive experiments to gather a database of any size for machine learning model evaluation, let alone training. Even small-scale psychophysical studies for stimuli of particular interest could prove very expensive.

Indeed, there are crucial scaling issues with the experimental procedure as it stands. The number of trials needed scales quadratically with the resolution of the desired segmentation mask, and multiplicatively with the number of objects. This is in stark contrast to traditional drawing methods which scale approximately linearly with both. I think this substantially limits the potential of the method for naturalistic images of any complexity. This limitation should be further expanded upon in the paper.

As such, even if the method/paper is modified to answer the questions mentioned in the previous two points, I see its value almost entirely in measuring the validity of more traditional segmentation methods. I don’t believe this issue is discussed in the paper, and I believe it should be, as it poses a strict limitation on the method’s use cases. That being said, such a tool could be tremendously useful, and if the previous two comments can be answered, I believe this paper could meaningfully contribute to our understanding of the potential biases (or lack thereof) in measuring segmentation using more traditional, scalable methods.

Minor comments

- Given that the per-image cost of segmenting is high, questions that relate to within-participant variability across stimuli seems difficult to assess. If one wants e.g. a database of 100 stimuli, this would effectively mean a single participant must perform 100 (or 800, if you want 5 objects!) sessions of the proposed experiment, which seems unreasonably difficult to accomplish in practice. This means that stimulus-specific variability and how that maps to perceptual variability (which is often what we are interested in as psychophysicists) can be difficult to measure using this methodology.

- Having to either fix the number of objects in the image beforehand, or having to collect a prohibitively large number of trials (in the paper, the example of a total of 4 hours of sessions per image per participant is given) appears to mean that one must curate the dataset of images to be segmented beforehand, and potentially impose large selection bias on the images that participants see. This could be especially problematic when one studies naturalistic images, where what counts as an independent object is often rather subjective and I worry that the method necessitates curating a dataset to exclude samples that could have many interpretations of the number of objects. This could lead to simpler or easier-to-segment images being selected for when making a benchmark, which could make human segmentation appear easier to explain using a model than it might really be.

- The use of MAE as a metric to judge segmentation mask error seems odd. In most of the literature I am familiar with, metrics like IoU/ARI are used. This makes it difficult to judge the simulations for how good the results are in general to quickly gauge the accuracy of the method. If I look at Figure 12, bottom-right 64x64 segmentation mask, even at MAE=0.050 the segmentation mask looks clearly very inaccurate and would certainly be dubbed as a ‘model failure’ in the ML literature. While I haven’t thought carefully about this, I wonder if comparing MAE across resolutions like this gives a meaningful measure of the ‘goodness’ of the segmentation mask.

Typos

- Author summary: “contributions are three folds” => “contributions are threefold”

- Line 134 and elsewhere in the document: “FIgure 6” => “Figure 6”

References

Russell, B.C., Torralba, A., Murphy, K.P. et al. LabelMe: A Database and Web-Based Tool for Image Annotation. Int J Comput Vis 77, 157–173 (2008). https://doi.org/10.1007/s11263-007-0090-8

M. Cordts, M. Omran, S. Ramos, T. Rehfeld, M. Enzweiler, R. Benenson, U. Franke, S. Roth, and B. Schiele, “The Cityscapes Dataset for Semantic Urban Scene Understanding,” in Proc. of the IEEE Conference on Computer Vision and Pattern Recognition (CVPR), 2016.

**Have the authors made all data and (if applicable) computational code underlying the findings in their manuscript fully available?**

Reviewer #1: **No: **

Reviewer #2: **No: **

PLOS authors have the option to publish the peer review history of their article (what does this mean?). If published, this will include your full peer review and any attached files.

Reviewer #1: No

Reviewer #2: No
---

## [Decision Letter · Decision Letter 1]

1 Aug 2023

Dear Prof. Vacher,

Thank you very much for submitting your manuscript "Measuring uncertainty in human visual segmentation" for consideration at PLOS Computational Biology. As with all papers reviewed by the journal, your manuscript was reviewed by members of the editorial board and by several independent reviewers. The reviewers appreciated the attention to an important topic. Based on the reviews, we are likely to accept this manuscript for publication, providing that you modify the manuscript according to the review recommendations.

Please address the remaining questions from R1

Sincerely,

Emma Claire Robinson

Academic Editor

PLOS Computational Biology

Daniele Marinazzo

Section Editor

PLOS Computational Biology

Reviewer's Responses to Questions

**Comments to the Authors:**

Reviewer #1: Re-review of "Measuring uncertainty in human visual segmentation"

Reviewer: Michael Landy

This version is certainly improved from the previous one and deals well with both my and the other reviewers' comments. Of course, last time I kvetched about math that I guessed the meaning of (generally correctly). It has been clarified, but on this reading I already knew what to expect ;^) I have very few comments at this point.

* 205: I^2 is set of all pairs, but you obviously don't use pairs of identical points, but at no point in the math do you point that out. Unimportant.

* 243: I'm still tripped up by the proposition. You state in the proposition that the family of p-vectors paired with any one fixed grid square should be linearly independent (i.e., span N-space). But, which vectors are these. This is a proposition about estimating those vectors. So, do you mean the ground-truth set of vectors are linearly independent. Or, do you mean that the p_i in Eq. 6 are linearly independent, i.e., the sets of p_i you try when you minimize (i.e., you only consider sets of {p_i} such that all grid squares' associated sets of emerging vectors are linearly independent? Unobvious from the notation, because you use the notation p_i to denote ground truth AND to denote the parameters over which you maximize likelihood of the data. This confusion holds for the proof in S1 as well (do you minimize only over sets that satisfy linear independence?).

* I like switching from independent to linearly independent in the text and would ask that you do so consistently. You missed 258, 286, 290, also the next to last line in S1

* 259: verified -> obtained???

* 315: problem -> Equation?

* 317: By "periodic" convolution do you mean one that wraps around the image edges? If so, why?

* 436: "Similar to" or "lower than" than that of a kernel? In fact, what do you mean by the dominant spatial frequency of a Gaussian, since that's a low-pass filter with dominant frequency that is always zero?

* S1: I don't understand the if-and-only-ifs in the proof. I'm sure it stems from linear independence, but couldn't work it out. You might clarify if you think that's appropriate.

* S2: Texture synthesis, line 3: There's a missing citation/latex bug

Reviewer #2: I believe that the authors' changes have made the manuscript stronger and congratulate them on an excellent manuscript. The authors have addressed all my comments and I fully support its publication

**Have the authors made all data and (if applicable) computational code underlying the findings in their manuscript fully available?**

Reviewer #1: Yes

Reviewer #2: None

PLOS authors have the option to publish the peer review history of their article (what does this mean?). If published, this will include your full peer review and any attached files.

Reviewer #1: **Yes: **Michael S Landy

Reviewer #2: No

Figure Files:

Data Requirements:

Reproducibility:

References:

---

## [Decision Letter · Decision Letter 2]

31 Aug 2023

Dear Prof. Vacher,

We are pleased to inform you that your manuscript 'Measuring uncertainty in human visual segmentation' has been provisionally accepted for publication in PLOS Computational Biology.

Best regards,

Emma Claire Robinson

Academic Editor

PLOS Computational Biology

Daniele Marinazzo

Section Editor

PLOS Computational Biology

Reviewer's Responses to Questions

**Comments to the Authors:**

Reviewer #1: It's ready for prime time now!

**Have the authors made all data and (if applicable) computational code underlying the findings in their manuscript fully available?**

Reviewer #1: Yes

PLOS authors have the option to publish the peer review history of their article (what does this mean?). If published, this will include your full peer review and any attached files.

Reviewer #1: No

---

## [Editor Report · Acceptance letter]

18 Sep 2023

PCOMPBIOL-D-23-00243R2 

Measuring uncertainty in human visual segmentation

Dear Dr Vacher,

I am pleased to inform you that your manuscript has been formally accepted for publication in PLOS Computational Biology. Your manuscript is now with our production department and you will be notified of the publication date in due course.

With kind regards,

Dorothy Lannert
